# Prematurity detection evaluating interaction between the skin of the newborn and light: protocol for the preemie-test multicentre clinical trial in Brazilian hospitals to validate a new medical device

Zilma Silveira Nogueira Reis,[1,2] Rodney Nascimento Guimarães,[2]
Maria Albertina Santiago Rego,[3] Roberta Maia de Castro Romanelli,[3]
Juliano de Souza Gaspar,[2] Gabriela Luiza Nogueira Vitral,[4]
Marconi Augusto Aguiar dos Reis,[3] Enrico Antônio Colósimo,[5]
Gabriela Silveira Neves,[6] Marynea Silva Vale,[7] Paulo de Jesus Hartamann Nader,[8]
Marta David Rocha de Moura,[9] Regina Amélia Pessoa Lopes de Aguiar[1]

For numbered affiliations see end of article.

**Correspondence to**
Professor Zilma Silveira Nogueira Reis;
zilma.medicina@gmail.com

## ABSTRACT

**Introduction** Recognising prematurity is critical in order to attend to immediate needs in childbirth settings, guiding the extent of medical care provided for newborns. A new medical device has been developed to carry out the preemie-test, an innovative approach to estimate gestational age (GA), based on the photobiological properties of the newborn's skin. First, this study will validate the preemie-test for GA estimation at birth and its accuracy to detect prematurity. Second, the study intends to associate the infant's skin reflectance with lung maturity, as well as evaluate safety, precision and usability of a new medical device to offer a suitable product for health professionals during childbirth and in neonatal care settings.

**Methods and analysis** Research protocol for diagnosis, singlegroup, singleblinding and singlearm multicenter clinical trial with a reference standard. Alive newborns, with 24 weeks or more of pregnancy age, will be enrolled during the first 24 hours of life. Sample size is 787 subjects. The primary outcome is the difference between the GA calculated by the photobiological neonatal skin assessment methodology and the GA calculated by the comparator antenatal ultrasound or reliable last menstrual period (LMP). Immediate complications caused by pulmonary immaturity during the first 72 hours of life will be associated with skin reflectance in a nested case–control study.

**Ethics and dissemination** Each local independent ethics review board approved the trial protocol. The authors intend to share the minimal anonymised dataset necessary to replicate study findings.

**Trial registration number** RBR-3f5bm5.

**Strengths and limitations of this study:**

► Prospective multicenter evaluation of a new medical device with training, and certification of collaborative centres.
► The gold standard comparator for pregnancy dating does not exist; instead a reference standard will be used with blinded primary outcome.
► The agreement endpoint between methods for gestational age determination precludes randomisation of the intervention.

to provide proper neonatal care. The day of birth is the riskiest for newborns and mothers almost everywhere.[1] Perinatal causes related to prematurity and complications during childbirth, which are generally preventable through qualified healthcare, are the primary causes of death among newborns.[1 2] Most of these deaths take place in countries with low resources and a scarcity of health facilities.[3] The opportune recognition of prematurity is critical in order to judge the viability of the newborn and to attend to his immediate needs, guiding the complexity of the medical care provided for the newborn. Without reliable information on the age of the unborn phase, actions to preserve the potential for survival of the newborn can be neglected.[4] Indeed, the attempted management of the risk of mortality and severe complications are sensitive issues to the gestational age (GA), which involves temperature maintenance,

## INTRODUCTION

In childbirth settings, health professionals continuously need to make timely decisions

ventilatory support, transport to a neonatal intensive care unit (NICU), and the early treatment of respiratory distress syndrome (RDS), the most severe complication of premature birth.[5] In addition to the GA information or birth weight, the prediction of neonatal respiratory morbidity may be critical in planning immediate medical care,[6] since the respiratory system is among the last of the fetal organ systems to mature, which is associated with enhanced morbidity and mortality.[6]

Current methods of dating pregnancy remain a worldwide challenge. Early obstetric ultrasound currently offers the best due date.[7] However, access to this type of exam is limited because of high equipment costs, poor training and skills of health professionals, or late prenatal care.[8] Despite a 10-days or more margin of error during the second and third trimester of gestation, ultrasound is still a reasonable methodology for GA determination, when the best opportunity was lost.[7] The calculation, based on the historical information of the last menstrual period (LMP), is impacted by the uncertainty of both the fertility days and date of conception,[9] due to the bias of memory, the use of hormonal contraception and breastfeeding.[10] After birth, neurological scores, such as the New Ballard,[11] show a tendency to overestimate GA in preterm infants and underestimate GA in growth-restricted infants.[12] Efforts to enhance the reliability of pregnancy dating, through more accurate and accessible technologies, seek to improve pregnancy outcomes and neonatal survival.[13]

A new medical device has been developed to carry out the preemie-test, an innovative approach used to estimate GA, based on the photobiological properties of the newborn's skin. This reflective test is non-invasive, and the device automatically processes the light, scattered by the constituents of the skin layers, when a small optoelectronic light emitter/receiver sensor touches the newborn's skin.[14] The device under test is easy to use and every effort is being made to ensure that it has excellent accuracy, be it safe and low cost. The feasibility study provided a mathematical model to predict GA based on the skin reflectance adjusted to clinical variables ($R^2$=0.828, p<0.001).[15] However, before the adoption or use of an innovation, an effectiveness trial of intervention is a critical step in the research chain regarding its the social utility when completing the translation from the proof of concept to clinical science.[15] The rationale for the main hypothesis in this study is that the skin maturity of a newborn, obtained by the analysis of its optical properties, is useful in pregnancy dating for clinical use and respiratory prognosis, especially in a scenario with no reliable GA based on current methods. This study aims to validate the photobiological model of the skin, called the 'preemie-test', in order to estimate GA at birth and determine its accuracy in detecting prematurity. Second, it also seeks to associate the infant's skin reflectance with lung maturity. Moreover, this study intends to evaluate the safety, precision and the usability of a new medical device to offer a suitable product to support health professionals during childbirth and in neonatal care settings.

## METHODS
### Study design
This study will use a protocol for diagnosis, singlegroup, singleblinding and singlearm multicenter clinical trial with a reference standard. This new photobiological approach to the skin, gathered in a medical device, is currently in the pivotal phase of innovation development from the prototype to regulatory approval.[16] This step aims to provide the translation[15] of the scientific model for GA detection based on skin maturity. This Protocol version is 2, 15 January 2019. Faculty of Medicine, Universidade Federal de Minas Gerais is the Coordinator Centre.

### Study settings, ethics and dissemination
Selected Brazilian referral centres for high-risk pregnancy and neonatal care will participate in the study, according to this protocol: Hospital das Clínicas, Universidade Federal de Minas Gerais, as the Centre for Coordination; Hospital Sofia Feldman, Minas Gerais State; Hospital da Universidade Luterana do Brasil, Rio Grande do Sul State; Hospital Materno-infantil de Brasília, Distrito Federal; and Hospital Universitário da Universidade Federal do Maranhão, Maranhão State. Each local independent ethics review board approved the trial protocol, and the Brazilian National Research Council approved all study activities and protocol prior to the commencement of study activities, in accordance with the Declaration of Helsinki (2008), good clinical practice as set forth by the International Organization for Standardization 14155:2011, and the Brazilian regulatory health agency's recommendations.[17] This study was logged under both protocol number CAAE 81347817.6.1001.5149 and the International Clinical Trials Registry Platform under number RBR-3f5bm5. Parents will sign an informed consent form on behalf of the newborn before participating in the clinical trial (online supplementary file).

### Patient and public involvement
Patients and the public were not involved in the design of this study. The results will be disseminated to study parents of participants through scientific publications, non-scientific publications and on the website of the project: http://skinage.medicina.ufmg.br.

### Eligibility criteria and participant's timeline
A prospective sequential and concurrent enrollment process will select newborns in referral hospitals centres for neonatal care. Infants are eligible with the following inclusion criteria: (1) alive newborn; (2) enrollment during first 24 hours of life; (3) be 24 weeks or more of GA, at birth; (4) fetus underwent an obstetric ultrasound assessment before 14 weeks of pregnancy and (5) fetus also had obstetric ultrasound assessment between 14 and 22 gestational weeks. Exclusion criteria are: (1) malformation with structural skin alterations and (2) skin modifiers: anhydramnios, hydrops, congenital skin diseases or chorioamnionitis. Randomisation was not appropriate to

assess the agreement between different methods to assess pregnancy dating.

In a nested case–control study, we will select newborns within the first 72 hours of life, discharge or death, whichever occurs first, with the following inclusion criteria: (1) RDS or (2) tachypnoea of the newborn (TTN) diagnosis. Ranges of GA will randomly pair controls. Exclusion criteria include: (1) the existence of extra pulmonary conditions with tachypnoea not due to prematurity and (2) diagnosis of Clinical or Laboratory-Confirmed Blood-stream Infection.

### Intervention: the preemie-test

The preemie-test assessment occurs as soon as possible after birth, in the first 24 hours, inside incubators, open heating crib, common crib or in the mother's lap, in order to ensure minimum manipulation and stable clinical conditions. The acquisitions of all newborns will be stored in a database for further statistical analysis.

A non-invasive, handheld optoelectronic prototype has been developed to measure the backscattered light signal from the skin.[15] The equipment regulates the emitted light and processes the received light signal in the sensor, resulting in the prediction of GA by a mathematical model, associated or not with clinical variables. According to the Brazilian regulatory health agency (ANVISA), this medical device is categorised as a Class II safety: non-invasive and medium risk. The prototype unit of measurement and the process of GA estimation were patented under number BR1020170235688 (CTIT-PN862).[14] An updated version of the invention received improvements in order to safeguard reliability and to minimise examiner interferences on the skin's back-scattering acquisition. The light emitting-sensor touches the skin over the sole of the foot for a few seconds. The skin reflectance will be sensed once the light has been emitted by a light emitting diode at wavelengths from 400 nm to 1200 nm. Data acquisitions occur automatically, without operator influence and are obtained three times per newborn, in the same site and sequentially. Digital recordings will be uploaded to a server for further analysis. The prototype will blind the examiner to the predicted GA value.

The criterium for discontinuing the interventions for a given trial participant will be in case of parents of the newborns' request.

### Training and monitoring

Systematic monitoring of data collection, through an electronic information system, would trigger any adverse event. This medical team is still responsible for the training of healthcare professionals to recruit participants, data collection, a safely performed preemie-test during the newborn's assessment and the monitoring of data quality. The certification of co-participant centres involved the accomplishment of at least 30 simulated examinations by the participant health professionals in the study.

### Gestational age methods of calculation and comparators

Reference-GA (R) is calculated on enrolment, using the embryo measurement assessed by ultrasound exam at <14 weeks of gestation as a reference. Crown-rump-length (CRL) data, recorded from the ultrasound report or prenatal care book-document, will be considered the crude data, when available. Intergrowth's 21st standard curve for ultrasound measurements from 7 weeks and 3 days up to 13 weeks and 6 days will be adjusted to all GA data, according to CRL.[18]

GA methods to calculate GA in the childbirth setting, and their comparators are as follows:
- ► Preemie-test-GA (T): data statistically determined by analysing the acquired information stored in the device's processor.
- ► Comparators-GA (C): calculated using the first ultrasound exam after 13 weeks and 6 days of gestation and before 22 weeks (C1). When available, a second comparator is GA based on a reliable LMP (C2).[13]

We will take a scanning copy of the prenatal care book or the ultrasound report. After evaluating the data quality, the images will be discarded. To achieve a reliable LMP, we will interview the woman, as suggested by Nguyen et al.[13]

### Primary outcome measures

The primary target is the agreement between the GA offered by the preemie-test (T) and the GA calculated by the comparators (C1 and C2), so as to perform the new test in scenarios without the Reference-GA (R). The outcome is the difference between the GA calculated by the photobiological neonatal skin assessment methodology in relation to the age calculated by the comparators.

Another measure for the primary target is the detection of preterm newborns, considering the age before 37 weeks of pregnancy as the threshold between term and preterm births, and analysing sub-categories of preterm birth, based on GA[4]:
- ► Extremely preterm (<28 weeks).
- ► Very preterm (28–32 weeks).
- ► Moderate to late preterm (more than 32 to <37 weeks).

In this case, the outcome is the proportion of the preterm newborn correctly detected at birth, based on the photobiological test of the skin, within a 1-week error.

### Secondary outcome measures

1. In a simulated scenario, in which the Reference-GA (R) is unknown, two groups will be randomly assigned from the complete database in order to compare differences among the Reference-GA (R), the GA obtained through the preemie-test (T) and the GA calculated by the comparators. Figure 1 presents such subgroups and measures for comparison.
2. To monitor the device's safety when in regular use by participants over a 72-hours period. Adverse events will be monitored, according to ISO 14155:2011 standards. This means any unexpected medical events, unintend-

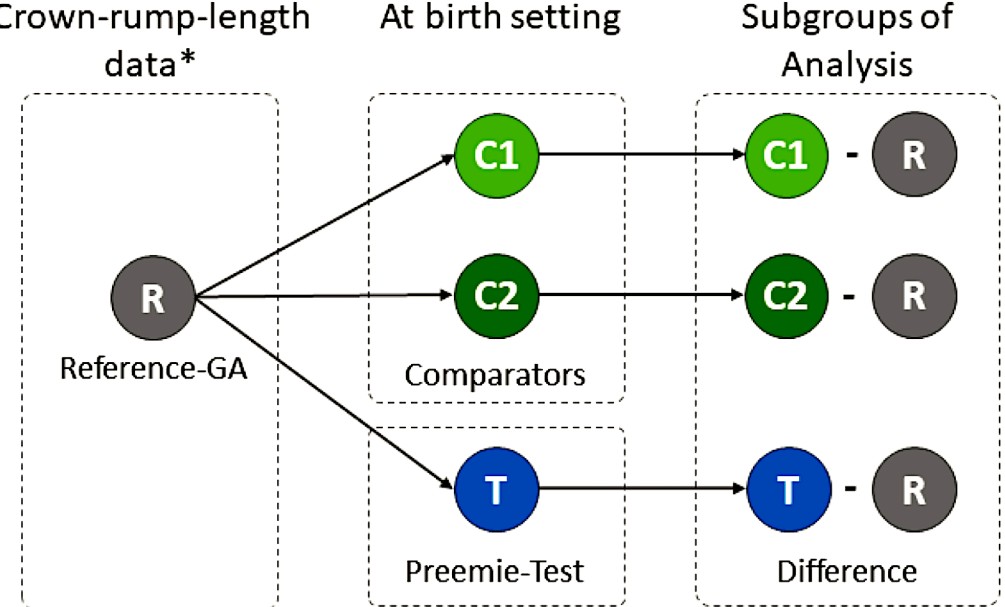

**Figure 1** Secondary outcome comparisons between the Reference-GA and the preemie-test in a simulated scenario without best pregnancy dating Legends: *Gestational age from crown-rump-length data adjusted to Intergrowth's 21st fetal standard.[18] R: reference. GA: gestational age. T, test; C1, comparator 1 is the GA calculated using the first ultrasound exam after 13 weeks and 6 days and before 22 weeks of gestation. C2, comparator 2 is the GA based on a reliable last menstrual period.

ed disease or injury or unfortunate clinical signs in subjects, users or other people, whether related to the investigational medical device or not.

3. To establish the *ease of use* of the preemie-test measurement as a potential method for preterm newborn diagnosis.

### The secondary outcome measures in the case–control nested study

Immediate complications, occurring during the first 72 hours of life due to pulmonary immaturity, are the secondary target. The outcome measures are as follows:

► To describe the relationship of the measurement of the newborn's skin reflectance with RDS and with diagnoses based on clinical and radiological findings and respiratory outcomes.[6 19]
► To describe the relationship of the measurement of the newborn's skin reflectance with the TTN and with diagnoses based on clinical findings and respiratory outcomes.[6]
► To describe the relationship of the measurement of the newborn's skin reflectance with ventilatory support due to pulmonary immaturity.
► To describe the relationship of the measurement of the newborn's skin reflectance with NICU admission due to RDS or TTN.

Time schedule of enrolment, intervention and outcome measurements are presented in a schematic diagram (see figure 2). The assessment occurs during the first 24 hours of life, but participants will be followed-up for 72 hours or until discharge or death, whichever occurs first, for the monitoring of neonatal outcomes and adverse events.

### Sampling and sample size

The sample size calculation is estimated based on the primary endpoint. To test the hypothesis of equivalence between the preemie-test GA and the comparators GA, a sample of 787 subjects is necessary to detect an effect size of 10%. Using the G-Power 3.1 software,[20] we assumed an alpha error of 0.05 and a power of test of 0.80 to support a paired t-test.

Sampling intends to arrange three groups of GA enrolment to preserve enough premature newborns with 3:2:1 proportion, similar to Wilson et al[21]: 392 term newborns, 263 premature newborns from 32 to 36 weeks and 6 days of GA and 132 extremely premature newborns from 24 to 31 weeks and 6 days of GA.

### Usability

The usability assessment will be performed by applying a checklist to participants who use the prototype device to perform the preemie-test. The 10 heuristics proposed by Nielsen and Marck (1994)[22] will be adapted to build a checklist to evaluate the device, namely: (a) system visibility, (b) correspondence with the real world, (c) user control and freedom, (d) consistency of results and standardisation, (e) error prevention, (f) visual recognition rather than memorization, (g) flexibility and efficiency of use, (h) aesthetic and minimalist design, (i) help for the user to recognise, diagnose and recover from errors and (j) user documentation and help.

| | STUDY PERIOD | | | |
|---|---|---|---|---|
| | Enrollment | Assessment | Close-out | Allocation |
| TIMEPOINT | 0 | 0 | 72 hours | Analysis |
| **ENROLLMENT:** | | | | |
| Eligibility screen | X | | | |
| Informed consent | X | | | |
| **INTERVENTION:** | | | | |
| Preemie-Test | | X | | |
| **ASSESSMENTS AND ANALYSIS:** | | | | |
| Preemie-Test: data acquisition | | X | | |
| Reference GA: calculated by obstetric ultrasound at <14 weeks of gestation | X | | | X |
| Comparator 1: GA calculated by obstetric ultrasound at ≥ 14 and <22 weeks | X | | | X |
| Comparator 2: GA calculated by reliable LMP | X | | | X |
| Case-control nested study: lung maturity | | ◆————————◆ | | |

**Figure 2** Participant timeline of the study GA, gestational age; LMP, last menstrual period; R, reference.

## Data collection

Standard operational procedures set data entries in structured questionaries. In this concurrent clinical trial, an electronic information system was developed to collect data in different hospitals, simultaneously. Entry forms validations were implemented with data values ranges to ensure the quality of the information. An audit of the data will be permanently performed and the data summary available on the project webpage. Double system, paper-based and electronic will permit audit concerning reliability and validity. Independent rater over-read all papers files and cross check with the electronic information.

## Data analysis

Demographics and baseline characteristics of the study group, as well the intervention measurements, will be summarised by the frequencies and the mean and standard deviation (SD), whereas the median and interquartile range (IQR) will be preferred for non-normally distributed continuous variables.

To model the GA prediction, computational randomisation will select two subsamples in the database. One of them to train the prediction model of GA based on skin reflectance and clinical variables, such as sex, time in an incubator, phototherapy, birth weight, among others. Another part will be for the analytical validation of the predictive model. Improvements in the existing prediction models for GA (preemie-test), will be conducted with conventional statistical and data mining analyses.

Regarding the primary endpoint, the agreement among three methods for GA will be calculated using the Intra-class coefficient correlation and Bland & Altman plots,[23] and paired t-testing. The accuracy of the preemie-test in identifying the premature newborn, within a 1-week margin of error, will be the target of the accuracy analysis.

The relationship between the measurement of the newborn's skin reflectance and complications due to pulmonary distress associated with immaturity will be evaluated by means of association tests and risk. The significance level for hypothesis tests will be 5%, together with 95% CIs.

## DISCUSSION
## Strengths and limitations

Availability of trustworthy GA information is a prerequisite for preterm birth classification and healthcare decisions.[24] In this light, the results of this clinical study have the potential to validate a new device for pregnancy dating. The preemie-test was prepared to operate with minimum operator intervention and for use by healthcare

professionals anywhere a birth takes place without a reliable GA .

The purpose of medical research involving neonates is intended to improve clinical procedures.[25] In this context, a clinical trial is a research study in which subjects are prospectively assigned to intervention and the effects of those interventions on health-related outcomes are thereby evaluated.[26] However, clinical trials on medical devices face barriers when an effective standard procedure does not exist, as is the case of the comparator procedure.[27] Our challenge in preparing the present protocol was the absence of a gold standard for pregnancy dating, since the fetal age begins on conception; however, this information is difficult to be accurately determined.[7] The study began with the training of health professionals in September 2018.

Planned Date of First Enrolment: 1 February 2019.

Planned Date of Last Enrolment: 31 Decembe 2019.

Data analysis will be finalised, the results of which are expected in May 2020.

**Author affiliations**
[1]Gynecology and Obstetrics, Universidade Federal de Minas Gerais, Faculty of Medicine, Belo Horizonte, Minas Gerais, Brazil
[2]Health Informatics Center, Faculty of Medicine, Universidade Federal de Minas Gerais, Belo Horizonte, Minas Gerais, Brazil
[3]Pediatrics, Faculty of Medicine, Universidade Federal de Minas Gerais, Belo Horizonte, Minas Gerais, Brazil
[4]Doctoral Pediatrics Postgraduate Program, Faculty of Medicine, Universidade Federal de Minas Gerais, Belo Horizonte, Minas Gerais, Brazil
[5]Statistics, Universidade Federal de Minas Gerais, Belo Horizonte, Minas Gerais, Brazil
[6]Hospital Sofia Feldman, Belo Horizonte, Minas Gerais, Brazil
[7]Pediatrics, Hospital Universitario da Universidade Federal do Maranhao, Sao Luis, Maranhão, Brazil
[8]Pediatrics, Universidade Luterana do Brasil. Hospital Universitário de Canoas, Porto Alegre, Rio Grande do Sul, Brazil
[9]Hospital Materno Infantil de Brasília, Brasília, Distrito Federal, Brazil

**Acknowledgements** The authors would like to thank Center for Clinical Research of Hospital das Clínicas, Universidade Federal de Minas Gerais for general support and Good clinical practices training; Fotini Toscas from the Brazilian Ministry of Health for the active intermediation as the trial sponsor contact.

**Contributors** ZSNR: designed the study, planned data collection, prepared the team for good clinical practices, wrote and revised the paper. RNG, RAPLA, MASR, RMCR and JdSG made substantial contributions to study design, planned data collection, prepared the team for good clinical practices, wrote and revised the paper. GLNV, MAAdR, GSN, PJN, MDRM and MSV made contributions to standard procedures in methods, drafted the manuscript, reviewed the paper and approved the final manuscript. EAC: drafted the work and reviewed it critically for important intellectual content, as statistic consultant.

**Funding** This research was supported by the Grand Challenges Exploration from the Bill & Melinda Gates Foundation (Grant number OPP1128907, Contract), http:// www.gatesfoundation.org/ and Fundação de Amparo a Pesquisa de Minas Gerais, Brazil, http://www.fapemig.br/en/,non-profitsectors. The clinical trial is funded by the Brazilian Ministry of Health, Program of Development of the Industrial Health Complex (PROCIS), project 23072.052747/2017-51, trial sponsor contact: Fotini Toscas, e-mail fotini.toscas@saude.gov.br. The funders played no role in the study design, data collection and analysis, decision to publish or preparation of the manuscript.

**Competing interests** Authors declare a patent deposit on behalf of the Universidade Federal de Minas Gerais and Fundação de Amparo a Pesquisa de Minas Gerais, Brazil, http://www.fapemig.br/en/. The inventors were Reis, Zilma Silveira Nogueira and Guimaraes, Rodney Nascimento: BR1020170235688 (CTIT-PN862).

**Patient consent for publication** Parental/guardian consent obtained

**Ethics approval** Roles and responsibilities: ZSNR is the Principal Investigator and coordinator of the Directive Committee. JSG is the coordinator of the Data Management Team and will continuously receive report adverse events of trial interventions or trial conduct. RAPLA is the coordinator of the Clinical Trial Quality Committee, responsible for important protocol modifications, if necessary.

**Provenance and peer review** Not commissioned; externally peer reviewed.

**Data sharing statement** The authors intend to share the minimal anonymized dataset necessary to replicate study findings. Data sharing will include: the reference and comparators GA, GA estimated by the Preemie-test, birth weight, RDS or transient tachypnea of the newborn (TTN) diagnosis, ventilatory support due to pulmonary immaturity, neonatal intensive care unit (NICU) admission due to RDS or TTN, and any adverse events regarding device's safety. Data and study-related documents as ethical approvals will be permanently accessible by URLs. The correspondent author, orcid.org/0000-0001-6374-9295, will provide data access under reasonable request since the original study citation is warranted.

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
