## [Reviewer comments · BMJ Open]

This paper was submitted to a another journal from BMJ but declined for publication following peer review. The authors addressed the reviewers' comments and submitted the revised paper to BMJ Open. The paper was subsequently accepted for publication at BMJ Open.

(This paper received three reviews from its previous journal but only two reviewers agreed to published their review.)

ARTICLE DETAILS

TITLE (PROVISIONAL)	Prematurity detection evaluating interaction between the skin of the newborn and light: Protocol for the Premie-Test multicenter clinical trial in Brazilian hospitals to validate a new medical device
AUTHORS	Reis, Zilma Silveira Nogueira; Guimarães, Rodney Nascimento; Rego, Maria Albertina Santiago; Maia de Castro Romanelli, Roberta; Gaspar, Juliano de Souza; Vitral, Gabriela Luiza Nogueira; dos Reis, Marconi Augusto Aguiar; Colósimo, Enrico Antônio; Neves, Gabriela Silveira; Vale, Marynea Silva; Nader, Paulo de Jesus Hartamann; de Moura, Martha David Rocha; de Aguiar, Regina Amélia Pessoa Lopes

VERSION 1 – REVIEW

REVIEWER	STEFANIA TRIUNFO High-Risk Obstetrical Unit Fondazione Policlinico Universitario 'A. Gemelli' IRCCS Università Cattolica del Sacro Cuore Rome, Italy
REVIEW RETURNED	06-Nov-2018

GENERAL COMMENTS	Zilma et co-workers propose a research protocol based on the prematurity detection evaluating interaction between newborn skin and light. By using the Premie-Test, a new medical device developed to estimate gestational age based on the photobiological properties of the newborn's skin. They aim not also to validate the Premie-Test for GA estimation at birth and its accuracy to detect prematurity, but also to associate the infant's skin reflectance with lung maturity, as well as evaluate safety, precision, and usability of the medical device, in order to offer a suitable product for health professionals during childbirth and in neonatal care settings. The research topic is of interest and requiring good investigations, with benefits for low and high-income countries. The study protocol is well designed and well-written. Additionally, it has been approved by prestigious Ethic Committes, as reported in the text, and it is ongonig from the past September. We wait their results.
--

REVIEWER	Everett Magann University of Arkansas for Medical Sciences Little Rock AR
REVIEW RETURNED	13-Nov-2018

GENERAL COMMENTS	The only other consideration might be to date pregnancies based on IVF where dating is most accurate and confirmed.
---

VERSION 1 – AUTHOR RESPONSE

Reviewer: 1 Reviewer Name: STEFANIA TRIUNFO Institution and Country: High-Risk Obstetrical Unit, Fondazione Policlinico Universitario 'A. Gemelli' IRCCS, Università Cattolica del Sacro Cuore, Rome, Italy Please state any competing interests or state 'None declared': NONE DECLARED

Please leave your comments for the authors below Zilma et co-workers propose a research protocol based on the prematurity detection evaluating interaction between newborn skin and light. By using the Premie-Test, a new medical device developed to estimate gestational age based on the photobiological properties of the newborn's skin. They aim not also to validate the Premie-Test for GA estimation at birth and its accuracy to detect prematurity, but also to associate the infant's skin reflectance with lung maturity, as well as evaluate safety, precision, and usability of the medical device, in order to offer a suitable product for health professionals during childbirth and in neonatal care settings.

The research topic is of interest and requiring good investigations, with benefits for low and high-income countries. The study protocol is well designed and well-written. Additionally, it has been approved by prestigious Ethic Committees, as reported in the text, and it is ongoing from the past September. We wait their results.

Reply to Reviewer 1: The authors thank you for the careful revision and your comments on the relevance of the study.

Reviewer: 2 Reviewer Name: Everett Magann Institution and Country: University of Arkansas for Medical Sciences Little Rock AR Please state any competing interests or state 'None declared': No competing interests Please leave your comments for the authors below The only other consideration might be to date pregnancies based on IVF where dating is most accurate and confirmed.

Reply to Reviewer 2

The authors thank you for the careful revision and your comments. We agree with you regarding a better pregnancy dating based on IVF procedures. We are planning this evaluation in the future. However, it spends an extended preparation and longer following of pregnant women. Right now, our intention is accelerating this innovation to low and medium income countries. **FORMATTING AMENDMENTS** (if any) Required amendments will be listed here; please include these changes in your revised version: - Please ensure that your **CORRESPONDING AUTHOR'S EMAIL ADDRESS** in your main document and Scholar One submission system are the same.

Reply: No, it was not. I changed it in the manuscript. - Please provide a better quality of your figure 2, ensuring the figure are not pixelated when zoomed in on.

Figures can be supplied in TIFF, JPG or PDF format (figures in DOCUMENT, EXCEL or POWERPOINT format will not be accepted), we also request that they have a resolution of at least 300 dpi and 90mm x 90mm of width.

Reply: Yes, we revised these specifications. - Please re-upload your supplementary file in PDF format.

Reply: Yes, we did it.